# Conjunctival Sac Microbiome in Infectious Conjunctivitis

**DOI:** 10.3390/microorganisms9102095

**Published:** 2021-10-04

**Authors:** Yasser Helmy Mohamed, Masafumi Uematsu, Yoshitomo Morinaga, Hien-Anh Thi Nguyen, Michiko Toizumi, Daisuke Sasaki, Katsunori Yanagihara, Duc-Anh Dang, Takashi Kitaoka, Lay-Myint Yoshida

**Affiliations:** 1Department of Ophthalmology and Visual Sciences, Graduate School of Biomedical Sciences, Nagasaki University, 1-7-1 Sakamoto, Nagasaki 852-8501, Japan; yasserhelmy@nagasaki-u.ac.jp (Y.H.M.); tkitaoka@nagasaki-u.ac.jp (T.K.); 2Department of Laboratory Medicine, Graduate School of Biomedical Sciences, Nagasaki University, Nagasaki 852-8501, Japan; morinaga@med.u-toyama.ac.jp (Y.M.); d-sasaki@nagasaki-u.ac.jp (D.S.); k-yanagi@nagasaki-u.ac.jp (K.Y.); 3National Institute of Hygiene and Epidemiology, Hanoi 100000, Vietnam; hienanh75@yahoo.com (H.-A.T.N.); dangducanhnihe@gmail.com (D.-A.D.); 4Department of Pediatric Infectious Diseases, Institute of Tropical Medicine, Nagasaki University, Nagasaki 852-8501, Japan; toizumi@nagasaki-u.ac.jp (M.T.); lmyoshi@nagasaki-u.ac.jp (L.-M.Y.)

**Keywords:** conjunctival microbiome, infectious conjunctivitis, polymerase chain reaction, 16S ribosomal DNA sequencing

## Abstract

Acute bacterial conjunctival infections are common, and this study identified the conjunctival bacterial community in infectious conjunctivitis cases seen at the outpatient clinic of Khanh Hoa General Hospital in Nha Trang, Vietnam from October 2016 through December 2017. Conjunctival swabs were collected and tested using conventional culture, PCR, and 16S ribosomal RNA sequencing. The study included 47 randomly selected patients. More than 98% of all DNA reads represented five bacterial phyla. Three of these phyla constituted 92% of all sequences (*Firmicutes* (35%), *Actinobacteria* (31%), and *Proteobacteria* (26%)). At the genus level, there were 12 common genera that constituted about 61% of all sequence reads. Seven of those genera were common (*Streptococcus* (10%), *Cutibacterium* (10%), *Staphylococcus* (7%), *Nocardioides* (7%), *Corynebacterium* 1 (5%), *Anoxybacillus* (5%), and *Acinetobacter* (5%)), which encompassed 49% of all reads. As for diversity analysis, there was no difference on PERMANOVA analysis (unweighted UniFrac) for sex (*p* = 0.087), chemosis (*p* = 0.064), and unclassified eyedrops (*p* = 0.431). There was a significant difference in cases with bilateral conjunctivitis (*p* = 0.017) and for using antibiotics (*p* = 0.020). Of the predominant phyla, *Firmicutes* had the highest abundance in bacterial conjunctivitis in this study. *Pseudomonas* as a resident commensal microbiota may have an important role in the prevention of infection.

## 1. Introduction

Acute bacterial conjunctival infections are common [1,2]. Although many cases show a benign course, some can be associated with sight-threatening ocular complications such as corneal ulcers, endophthalmitis, panophthalmitis, and perforation of the globe [3]. Identification of the causative pathogens in these cases is mandatory but often difficult because some bacteria have special growth requirements [4]. The causative pathogens of bacterial conjunctivitis have been determined conventionally by both the smear method and the culture method. However, the results of these methods are not always conclusive [5]. The bacterial species detected in eyes with bacterial conjunctivitis have also been found in normal conjunctival sacs [6,7]. Furthermore, the sample size from ocular tissues is usually small, leading to unreliable cultivation results, and it is not easy to detect the rarely encountered, slowly growing, and uncultivatable bacteria. Initiation of proper therapy can be delayed with possible devastating visual consequences [4].

Because of these limitations, the advent of cultivation-independent techniques of microbial identification, such as polymerase chain reaction (PCR) and 16S ribosomal RNA (rRNA) sequencing, has provided a much more detailed picture of the human bacterial microbial consortium than was available through traditional culture techniques [8].

In 2011, a first pilot study involving four normal subjects was conducted with an aim to explore the bacterial diversity of a healthy human conjunctiva using 16S rRNA sequencing [9]. The study revealed an unpredicted diverse microbial community and identified that healthy conjunctival microbiome was dominated by *Proteobacteria*, *Actinobacteria* and *Firmicutes* bacterial phyla. The most common taxa at the genus level were *Pseudomonas*, *Propionibacterium*, *Bradyrhizobium*, *Corynebacterium*, *Acinetobacter*, *Brevundimonas*, *Staphylococci*, *Aquabacterium*, *Sphingomonas*, *Streptococcus*, *Streptophyta* and *Methylobacterium* [9]. In 2016, using the same technology to analyze 31 normal conjunctival samples, Huang et al. identified a high microbial diversity, classified into 25 phyla and 526 distinct genera, providing a framework to investigate the potential roles played by diverse microbiota [10]. In terms of composition, these studies concluded that *Proteobacteria* was the most abundant phylum on the normal conjunctiva while *Actinobacteria* and *Firmicutes* were the next two most abundant phyla [9,10].

Thus, the purpose of this study was to identify the bacterial community in conjunctival sacs of eyes with infectious conjunctivitis by using 16S rRNA metagenome sequencing.

## 2. Patients and Methods

### 2.1. Study Design

This was a prospective, observational study at the outpatient clinic of the Department of Ophthalmology, Khanh Hoa General Hospital in Nha Trang (only public referral hospital in Nha Trang city), central Vietnam. The institutional review committees of the hospital approved the protocol of this study, which adhered to the tents of the Declaration of Helsinki. Clinical-epidemiological data were collected and informed consent was obtained before beginning the examinations and collection of the samples. Clinical-epidemiological data included age, sex, laterality of infection, associated chemosis, pain, itching, visual loss, usage of unclassified eye drops, and antibiotic eye drops.

### 2.2. Sample Collection and Testing

Patients of any age with infectious conjunctivitis visiting the ophthalmology outpatient department were enrolled if they provided their written, informed consent, from October 2016 through December 2017 for bacterial community identification. Conjunctival swabs were carefully obtained from the more infected patient’s eye. There were 793 conjunctivitis cases enrolled in the conjunctivitis study during that study period. Due to budget limitations, 50 cases were randomly selected among these samples using a random list generated by a computer. The randomly selected samples were used for detection of micro-organisms using conventional culture and PCR examinations.

Tubes with 800 µL of normal saline were prepared, and their caps were opened before collecting the sample by soaking the swab with 2–3 drops of sterile saline, pulling down the lower eyelid of the most inflamed eye, and gently sweeping the swab on the conjunctiva from inner to outer canthus. The swab was placed in the medium to the bottom, the shaft of the swab was cut by sterilized scissors, the medium’s screw top was closed, and the specimen was kept at 4 °C until analyzed.

Specimens were collected and the initial conventional cultures (blood and a chocolate agar nutrient medium) were conducted at Khanh Hoa General Hospital same day of sampling. Then ocular samples were stored at −20 °C until transferred to our hospital. DNA was extracted and screened by real-time PCR and 16S rRNA sequencing assay in the Department of Laboratory Medicine at Nagasaki University Hospital on March 2018.

### 2.3. Polymerase Chain Reaction (PCR) Amplification and Preparation for 16S rRNA Gene Sequencing

Data of the 16S rRNA metagenome were obtained as described previously by Morinaga et al. [11]. DNA was extracted using a Quick-DNA Fecal/Soil Microbe Miniprep Kit (ZYMO Research, Irvine, CA, USA), according to the manufacturer’s instructions. The V1-V2 region of the bacterial 16S rRNA genes was amplified. After emulsion PCR, enriched samples were loaded onto an Ion 318 chip, and sequencing was performed using the Ion Torrent Personal Genome Analyzer (Thermo Fisher Scientific, Waltham, MA, USA).

### 2.4. Sequence Analysis

The sequencing reads were analyzed using CLC Genomics Workbench version 12.0.1 and CLC Microbial Genomics Module version 3.6.11 (QIAGEN N. V., Venlo, The Netherlands), as described previously [11]. After removing the primer sequences and trimming the read length between 200 bp and 400 bp under a 0.01% quality limit, samples with fewer than 100 reads and less than 50% from the median were excluded from further analyses. Chimeric reads were filtered using the chimera crossover detection algorithm with the default parameters. The reads were categorized into operational taxonomic units (OTUs) with 97% similarity and then assigned using SILVA release 132. The number of OTUs, the Shannon index (alpha diversity), and weighted UniFrac distances were analyzed using the CLC Microbial Genomics Module. Differences in bacteria abundance were calculated using LEfSe with default parameters as described by Segata et al. [12]. Differences at the genus level of bacterial abundance were analyzed after removing sequences of mitochondria and non-available ones from the data. We provide PCR primers as a Appendix A.

### 2.5. Statistical Analysis

To compare beta diversity, the data were analyzed by PERMANOVA analysis using the CLC, and a significant alpha level was set as *p* ≤ 0.05 (Mann–Whitney test) for the false discovery rate (FDR).

## 3. Results

The final cases included 47 randomly selected infectious conjunctivitis cases, because three samples were discarded due to a technical fault. Twenty-eight female and 19 male patients were included in this study, with an average age of 38.0 ± 20.6 (range 1–74) years (Table 1).

### 3.1. Bacterial Community Composition in Conjunctivitis

To identify bacterial composition in human conjunctivitis, the 16S rRNA metagenomic sequences were classified at both the phylum and genus levels (Figure 1 and Figure 2). More than 98% of all DNA reads represented five bacterial phyla. Three of these phyla constituted 92% of all sequences (*Firmicutes* (35%), *Actinobacteria* (31%), and *Proteobacteria* (26%)) (Figure 1). The other two phyla (*Bacteroides* (6%) and *Cyanobacteria* (0.9%)) were present in lower quantities. *Acidobacteria*, *Fusobacteria*, and others were present in contamination-level quantities (0.5% or less).

At the genus level, there were 12 common genera that constituted about 61% of all the sequence reads (Figure 2). Seven of those genera were common (*Streptococcus* (10%), *Cutibacterium* (10%), *Staphylococcus* (7%), *Nocardioides* (7%), *Corynebacterium* 1 (5%), *Anoxybacillus* (5%), and *Acinetobacter* (5%)), which encompassed 49% of all reads. The other five were less abundant (*Janibacter* (3%), *Porphyromonas* (3%), *Bacillus* (2%), *Clostridium sensu stricto* 7 (2%), and *Haemophilus* (2%)).

### 3.2. Diversity Analysis

As for diversity analysis, there were no differences on PERMANOVA analysis (unweighted UniFrac) (Beta diversity analysis) for sex (*p* = 0.087), chemosis (*p* = 0.064), and unclassified eye drops (*p* = 0.431). There were also no differences on PERMANOVA analysis for pain (*p* = 0.315), itching (*p* = 0.133), and visual loss (*p* = 0.05005). There were significant differences in bilateral versus unilateral conjunctivitis (*p* = 0.017) (Figure 3) and for using antibiotics (*p* = 0.020) (Figure 4) (beta diversity analysis), although there were no significant differences in these factors using the Shannon index (alpha diversity analysis).

### 3.3. Relative Abundance of Each Sample

Next, the relative abundance of each sample at phylum and genus levels were analyzed and compared based on the information about the use of antibiotics and laterality. In the analysis of the phylum level, significant findings were not observed between the groups for the use of antibiotics (Figure 5A), however, for the laterality, samples in the bilateral group seemed to be relatively rich in *Firmicutes* rather than the unilateral group (Figure 5B). In the analysis of representative genera (which had at least 50,000 reads in this study), *Streptococcus, Cutibacterium*, and *Staphylococcus* genera were observed in most of samples. Samples with unique genera with high abundance, such as *Porphylomonas, Haemophilus, Corynebacterium, Acinetobacter, Janibacter,* and *Anaerococcus* were also observed (Figure 5C,D).

### 3.4. Linear Discriminant Analysis

Linear discriminant analysis (LDA) was also performed to identify significant bacteria, which were associated with laterality and the use of antibiotics.

No significant phylum was observed in either group (maybe due to large variation) (Figure 6A).

The genera *Prevotella*, *Pyrinomonas*, *Tessaracoccus*, *Pelomonas*, *Exigubacterium*, and *Roseibacterium* were increased in the non-antibiotic group, and the genera *Paenibacillus*, *Afipia,* and *uncultured*_27 were increased in the antibiotic group (Figure 6B).

The phyla *Bacteroidetes* and *Acidobacteria* were significantly increased in the unilateral group, whereas no significant phylum was observed in the bilateral group (Figure 6C, D).

*Porphyromonas*, *Prevotella*, and *Stenotrophomonas* genera were increased in the unilateral infective group, and genera *Cutibacterium*, *Dolosigranuulum, uncultured*_29, and *Clostridium sensu stricto 12* were increased in the bilateral group (Figure 6E).

### 3.5. Comparison of Cultivation and Metagenome

A considerable difference was observed in the increased diversity of bacterial populations determined by rRNA sequencing compared with cultivation. Cultures were only positive in four cases (8.5%). Three of them were positive for Gram-positive cocci and the fourth case was positive for Alpha-hemolytic *Streptococci*. One of the three cases was also positive for *Acinetobacter* spp.

In case No. 401, the most abundant bacterium in the metagenome result was *Acinetobacter* (38%), which was congruent with the conjunctival cultivation result (Figure 7). In the other three cases (No. 008, 434, 487), the most abundant bacteria were *Streptococci* and *Staphylococci* in the metagenome, which were also congruent with the conjunctival cultivation results (Figure 7).

## 4. Discussion

### 4.1. Overview of the Microbiome in All Patients

A culture-independent approach by 16S rRNA metagenome sequencing has been used to identify our microbiota during both health and illness.

In a smaller study of only four healthy volunteers, Dong et al. identified a core conjunctival microbiome of five bacterial phyla using 16S rRNA sequencing, three of which (*Proteobacteria* (64%), *Actinobacteria* (19.6%), and *Firmicutes* (3.9%)) accounted for > 87.9% of all sequences [9]. The other two phyla, *Cyanobacteria* and *Bacteroidetes,* were found in contamination-level quantities (0.21% and 0.16%, respectively) [9]. In another larger study (31 subjects), Huang et al., using 16S rRNA gene sequencing reads, classified the conjunctival microbiome into 25 bacterial phyla. Most sequences (98.88%) were affiliated predominantly with five phyla, which included *Proteobacteria* (46.50%), *Actinobacteria* (33.89%), *Firmicutes* (15.50%), *Bacteroidetes* (2.28%), and *Deinococcus-Thermus* (0.71%) [10].

Although *Firmicutes* was the least abundant in their studies among the main first three phyla (*Proteobacteria, Actinobacteria, Firmicutes)*, in the present study it had the highest abundance (one of the most significant results in this study).

In another study, the conjunctivae of 45 healthy subjects were sampled at three time points over three months with the aim of understanding whether the microbial communities of the ocular surface (OS) change over time. They determined that the majority of phyla at each time point consisted of *Proteobacteria* (range 52–73%), *Firmicutes* (13–20%), and *Actinobacteria* (8–22%) [13]. Thus, the present study is the first to show that *Firmicutes* is the predominant phylum in the conjunctival microbiome in conjunctivitis cases. It is difficult to explain the reason why the phylum *Firmicutes* is predominant in patients with conjunctivitis. However, the consumption of oxygen by growth of causative bacteria including facultative aerobes may provide an anaerobic condition, which has the advantage for the phylum *Firmicutes.*

In a comparison between the present study and other previous normal studies, as regards to the predominant genera (>1%), there were three groups in the present study. The first group included the predominant genera shared with other normal studies [9,10,13,14]. This group included *Streptococcus*, *Cutibacterium, Staphylococcus*, *Corynebacterium*, and *Acinetobacter*. The second group included genera only present in the current study and not present in previous normal studies. This group included *Nocardioides, Anoxybacillus, Janibacter*, *Porphyromonas*, *Bacillus*, *Clostridium sensu stricto* 7, and *Haemophilus*. The last group included the predominant genus that is present in normal studies and absent in the current study, *Pseudomonas* [9,10,13,15].

Advances in next-generation sequencing and bioinformatics tools have shown an expansive and diverse microbial community inhabiting the human conjunctiva. The most abundant genera identified using 16S rRNA sequencing were *Pseudomonas*, *Bradyrhizobium*, *Cutibacterium*, *Acinetobacter*, and *Corynebacterium* [15,16].

It has been suggested that these “normal bacteria” serve a protective role under most circumstances by directly inhibiting colonization of more pathogenic species [17]. Many previous studies confirmed that *Pseudomonas* represented a major genus in healthy conjunctiva [9,10,13]. Lee et al. compared ocular microbial communities with and without blepharitis [18]. They confirmed that the relative proportions of *Staphylococcus*, *Streptophyta*, *Corynebacterium*, and *Enhydrobacter* were higher in subjects with blepharitis than in healthy subjects [18]. However, the proportion of *Pseudomonas* was clearly lower in subjects with blepharitis than in healthy subjects, suggesting that *Pseudomonas* might be important as a resident commensal microbiota for the prevention of blepharitis [18].

In the present study, it was confirmed that *Pseudomonas* was present in a scant percentage of cases of conjunctivitis, which may agree with the previous study in their suggestion about the importance of *Pseudomonas* as resident commensal microbiota for prevention of infection. Further study is needed to confirm this conclusion with own control specimens.

For the ocular microbiota during purulent conjunctivitis, studies using previous methods such as denaturing gradient gel electrophoresis (DGGE) and clone library methods have shown that the genera *Staphylococcus*, *Corynebacterium*, and *Cutibacterium* were commonly observed [19,20]. However, there is a bias in these studies because the findings were based primarily on the database of clinically-known bacteria. Thus, the present study provided the composition of ocular microbiota including uncultured bacteria with a reduction in the methodological selection bias.

### 4.2. Microbial Diversity

The diversity of microbial communities in the subjects was assessed with alpha diversity analysis. The observed genesis and the Shannon index were used to evaluate the richness and biodiversity of the microbiota. Beta diversity refers to species diversity among different groups. Beta diversity and alpha diversity together constitute the biological heterogeneity of overall diversity or a certain community or group. The beta diversities in different groups were calculated using the Unweighted UniFrac distances of 16S rRNA genes between microbial communities or groups.

In the present study, there were no differences on beta diversity analysis for sex (*p* = 0.087), chemosis (*p* = 0.064), unclassified eye drops (*p* = 0.431), pain (*p* = 0.315), visual loss (*p* = 0.05005), and itching (*p* = 0.133). There were significant differences between bilateral versus unilateral conjunctivitis (*p* = 0.017) and for antibiotics (*p* = 0.020) (Figure 3 and Figure 4), although there were no significant differences in these factors using the Shannon index (alpha diversity analysis). These findings suggested that the bacterial composition of the microbiota was changed without reducing the species diversity. Alteration of the microbiota due to antibiotics usage was characterized with the reduction of some susceptible bacteria and the induction of microbial substitution. In addition, it was confirmed that antibiotic administered prophylactically before ophthalmic procedures may further reduce the ocular surface’s amount and variety of microbiota [21,22]. Reasonably, we can accept that bacterial composition of the microbiota in bilateral cases was different from unilateral cases. However, the stability in the alpha diversity can be a characteristic of the ocular microbiota in contrast to skin microbiota [23]. These results, to the best of our knowledge, are considered to be the first diversity analyses for subjects with conjunctivitis reported in the literature.

### 4.3. Linear Discrimination Analysis

LDA was also performed to identify significant parameters from large data and found that Phyla *Bacteroidetes* and *Acidobacteria* were significantly increased in the unilaterally infected group, whereas no significant phylum was observed in the bilaterally infected group. In addition, *Porphyromonas*, *Prevotella,* and *Stenotrophomonas* genera were significantly increased in the unilateral group. These results were reasonable, because genera *Porphyromonas* and *Prevotella* actually belong to the phylum *Bacteroidetes* and would support the present results.

### 4.4. Individual Metagenomics

The rate for the culture method has been reported to be between 47.5% and 97.8% in eyes with bacterial conjunctivitis [19,20,24] and 9.0% to 90.6% [6,7,10,25,26,27] in normal conjunctival sacs.

To overcome culture limitations, there has been an increase in the number of studies using molecular methods, for example, PCR with species-specific primers [28,29,30], amplification of the 16S rRNA gene by PCR using universal primer sets followed by direct sequencing [31,32,33], DGGE [19], and pyrosequencing [9].

The 16S rRNA sequencing has been used for bacterial identification and discovery of novel genera, leading to extending our knowledge about OS microbial diversity [4,9,10,14].

In the present study, there were only 4 (8.5%) positive conjunctival cultures, and all of them were congruent with abundant genera on 16S rRNA sequencing analysis. From our point of view, if the result of 16S rRNA sequencing analysis shows an abundance of one or more genera in the conjunctivitis sample, these are most probably the causative pathogen(s), especially if the culture is congruent.

### 4.5. Limitations

There were some limitations to our study. First, there were no control normal conjunctival samples. Second, the 16S rRNA sequencing analysis method is unable to identify bacterial species level and is prone to noise, sampling errors, and contamination. Finally, using metagenomics data, extensive results that generate huge amounts of data are produced, which becomes difficult for a clinician to record and analyze on a routine basis. Despite these limitations, our data add to the growing understanding of the conjunctivitis microbiome.

## 5. Conclusions

Of the predominant phyla, *Firmicutes* had the highest abundance in bacterial conjunctivitis in this study. *Pseudomonas* as a resident commensal microbiota may have an important role in the prevention of infection.

## Figures and Tables

**Figure 1 microorganisms-09-02095-f001:**
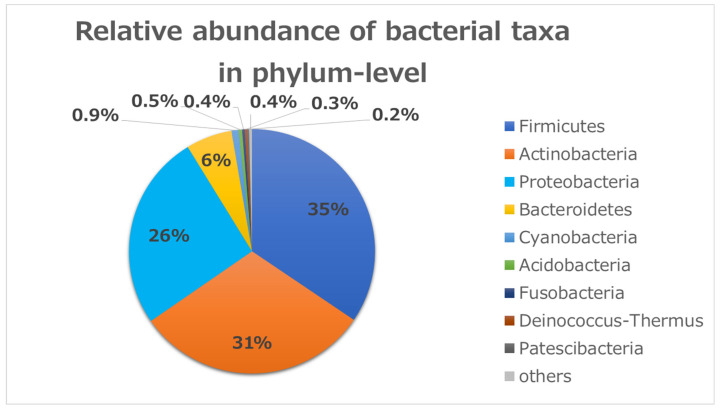
Relative bacterial compositions of conjunctivitis samples. 16S rRNA gene sequences are classified into phylum levels.

**Figure 2 microorganisms-09-02095-f002:**
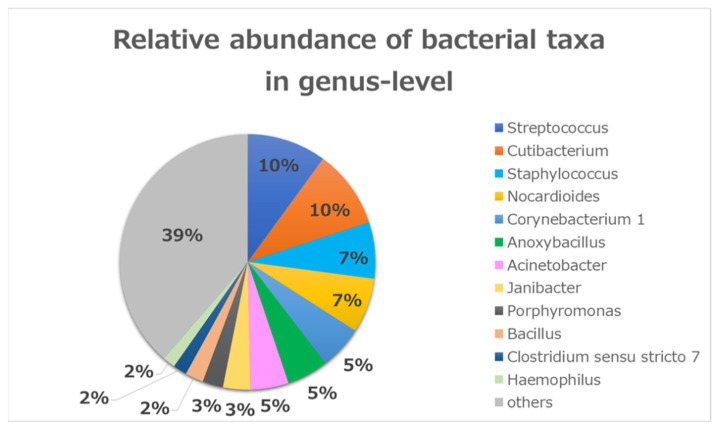
Relative bacterial compositions of conjunctivitis samples. 16S rRNA gene sequences are classified into genus levels.

**Figure 3 microorganisms-09-02095-f003:**
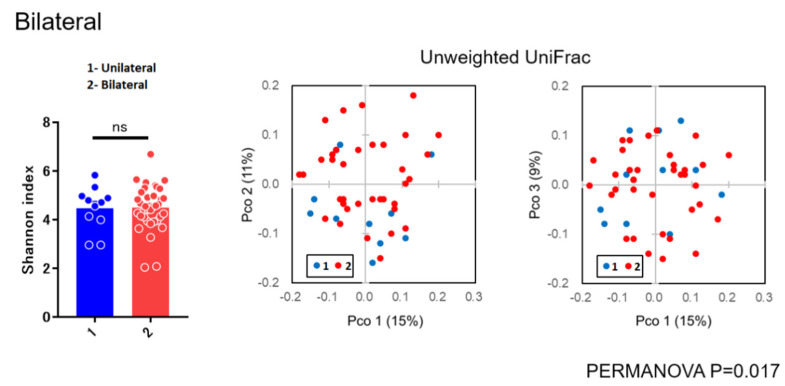
Alpha and beta diversity analysis for unilateral versus bilateral conjunctivitis.

**Figure 4 microorganisms-09-02095-f004:**
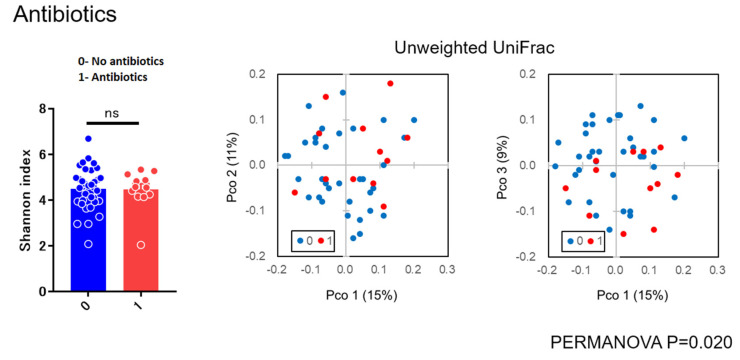
Alpha and beta diversity analysis for group not using versus using antibiotic eye drops.

**Figure 5 microorganisms-09-02095-f005:**
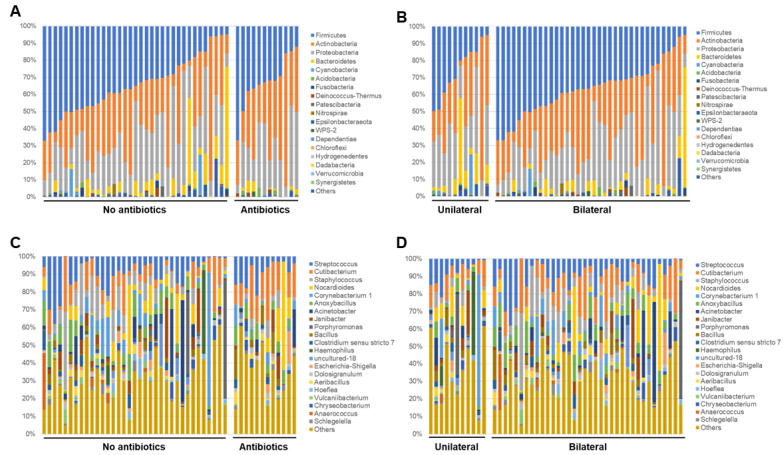
Relative abundance of each sample at phylum (**A**), (**B**) and genus (**C**), (**D**) levels as regards use of antibiotics and laterality.

**Figure 6 microorganisms-09-02095-f006:**
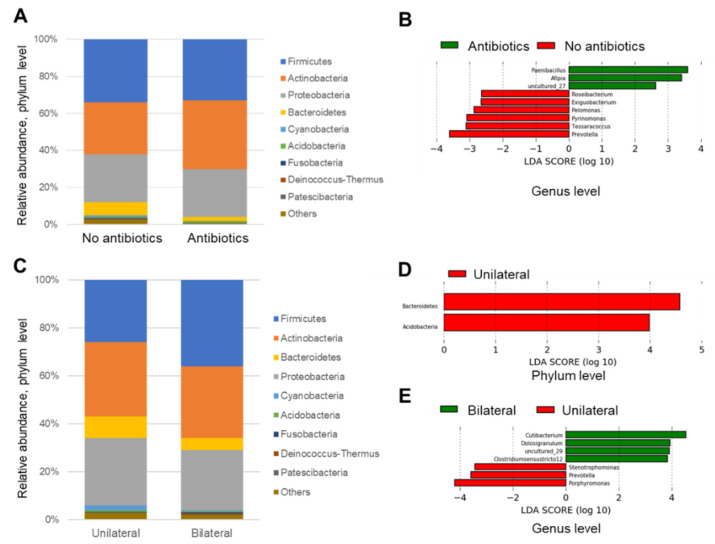
Linear discriminant analysis for antibiotic and bilateral groups (**A–E**).

**Figure 7 microorganisms-09-02095-f007:**
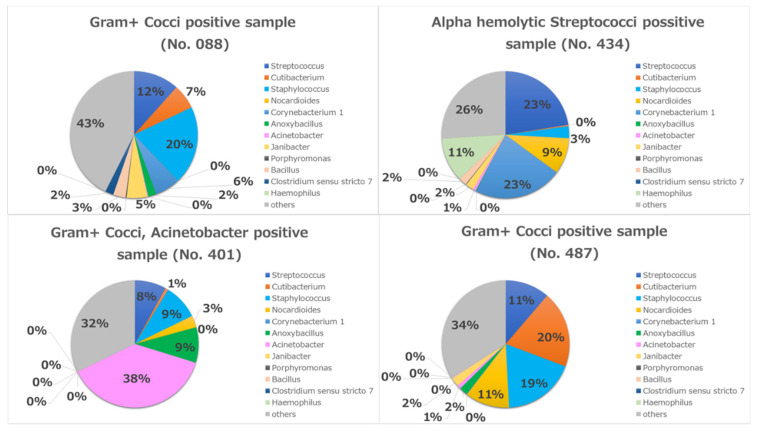
Individual metagenomics in four culture-positive cases.

**Table 1 microorganisms-09-02095-t001:** Table showed the detailed metadata of all patients included.

ID	Age	Sex	Uni/Bilateral	Antibiotics Drops	Sample Collection Date	Underlying Medical Condition	Culture
12	67	Male	Bilateral	-	10/25/2016		
17	20	Male	Bilateral	+	10/25/2016		
19	1	Female	Bilateral	-	10/25/2016		
51	40	Female	Unilateral	+	11/7/2016		
60	57	Female	Unilateral	-	11/11/2016		
64	53	Male	Bilateral	-	11/17/2016	Post-cataract surgery and IOL	
67	68	Female	Bilateral	+	11/17/2016		
75	33	Male	Bilateral	-	11/25/2016		
88	46	Female	Bilateral	-	12/2/2016		Gram+ *Cocci*
92	1	Female	Unilateral	+	12/5/2016		
94	26	Female	Bilateral	-	12/6/2016		
111	71	Female	Bilateral	-	12/20/2016	Hypertension	
117	49	Female	Bilateral	+	12/20/2016	Hypertension	
123	8	Female	Bilateral	-	12/21/2016		
149	40	Male	Bilateral	+	1/12/2017		
163	19	Male	Bilateral	+	2/13/2017		
165	33	Male	Bilateral	+	2/13/2017		
173	56	Female	Bilateral	-	2/23/2017	Diabetes mellitus	
174	22	Male	Bilateral	+	2/23/2017		
177	26	Male	Bilateral	+	2/28/2017	Dyslipidemia	
194	20	Male	Bilateral	-	3/16/2017		
195	23	Female	Unilateral	-	3/17/2017		
236	48	Male	Bilateral	-	3/31/2017	Smoking	
255	59	Female	Bilateral	-	4/14/2017	Hypertension	
278	35	Female	Bilateral	-	4/28/2017		
284	43	Female	Bilateral	-	5/8/2017		
317	8	Female	Bilateral	-	5/15/2017		
339	34	Female	Bilateral	-	5/23/2017		
347	6	Female	Bilateral	-	5/29/2017		
362	73	Male	Unilateral	-	6/13/2017	Cataract	
367	29	Female	Unilateral	-	6/13/2017		
373	49	Male	Bilateral	-	6/22/2017	Hypertension	
376	46	Female	Bilateral	-	6/23/2017		
401	65	Male	Bilateral	-	7/17/2017	Smoking	Gram+ *Cocci, Acinetobacter*
410	55	Female	Unilateral	-	7/28/2017		
422	45	Male	Bilateral	-	8/7/2017		
424	16	Male	Unilateral	-	8/8/2017		
434	71	Female	Unilateral	-	8/24/2017	Hypertension	*Alfa hemolytic Streptococci*
446	35	Female	Bilateral	-	8/28/2017		
455	45	Female	Unilateral	-	9/7/2017		
461	40	Female	Bilateral	-	9/11/2017		
463	17	Female	Bilateral	+	9/11/2017		
479	3	Male	Unilateral	-	9/14/2017		
484	28	Male	Bilateral	+	9/21/2017		
487	35	Female	Bilateral	-	9/22/2017		Gram+ *Cocci*
507	74	Male	Bilateral	-	9/29/2017		
530	66	Female	Bilateral	-	10/10/2017		

## Data Availability

The datasets generated and/or analysed during the current study are available with the corresponding author and ready to submit when needed.

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
