# Peer review of "Conjunctival Sac Microbiome in Infectious Conjunctivitis"

_microorganisms, 2021, doi:10.3390/microorganisms9102095_

Round 1

Reviewer 1 Report

I would like to thank the authors for answer addressing all of my concerns. The manuscript would be strengthened by additional experiments including controls; however, the authors acknowledge the limitation in the text and the information is still important for researchers and clinicians to have available.

Reviewer 2 Report

Authors answered all questions raised by the reviewers. 

This manuscript is a resubmission of an earlier submission. The following is a list of the peer review reports and author responses from that submission.

Round 1

Reviewer 1 Report

Thank you for the opportunity to review this manuscript.

Yasser et. al., investigate the bacterial community diversity among the infectious conjunctivitis cases using Ion Torrent Personal Genome Analyzer of 16S rRNA V1-V2 region and traditional culture methods. The authors focused on the randomly selected 47 patients samples and missed the normal control samples and swab control which is one of the critical in the element in the metagenomic analysis. The issues have been discussed by the author but not convincing. In this study, the authors reported three major phyla were present in 92% of all sequences. Further concluded that Firmicutes are the predominant phyla in bacterial conjunctivitis. I have some specific comments in the following that the authors may consider in further improving their manuscript.

Specific comments:

  • Publication related to conjunctival microbiota shown in the discussion should also emphasis in the introduction.
  • The authors have been described the sample collection and selection. Provide some detail of durations between sampling and samples processing
  • Provide detailed metadata including Age, sex, uni/bilateral, with/without antibiotic, date of sample collection, processing info – immediately or stored for sample pool.
  • Could the authors comment on their choice to use the OTU approach rather than the recently more commonly used ASV approach?
  • The authors have provided the Shannon index for alpha diversity, however would like to see the relative abundance of the bacterial diversity for each sample. So I would suggest the author provide the relative abundance (Figure 5 A and C) of all 47 samples (both genus and phylum level) and discuss how the evenness and richness of the diversity using the metadata.
  • No detailed report on the Beta diversity authors should provide more detail on this aspect.
  • Figure 6 color pattern unable to distinguish so use different colors.
  • Authors claiming this is the first study trying to identify bacterial composition in human conjunctivitis but few investigations have been listed by authors and few pieces of literature available. I would suggest rephrasing or remove the sentence.

Author Response

18-June-2021

Thank you very much for your comments on our manuscript.

We have revised the manuscript as per your suggestion.

Reviewer: #1

Comments to the Author:

  • Publication related to conjunctival microbiota shown in the discussion should also emphasis in the introduction.

We added the following sentences in Introduction as per your suggestion.

“In 2011, a first pilot study involving 4 normal subjects was conducted with an aim to explore the bacterial diversity of a healthy human conjunctiva using 16S rRNA sequencing [9]. The study revealed an unpredicted diverse microbial community and identified that healthy conjunctival microbiome was dominated by Proteobacteria, Actinobacteria and Firmicutes bacterial phyla. The most common taxa at the genus level were Pseudomonas, Propionibacterium, Bradyrhizobium, Corynebacterium, Acinetobacter, Brevundimonas, Staphylococci, Aquabacterium, Sphingomonas, Streptococcus, Streptophyta and Methylobacterium [9]. In 2016, using the same technology to analyze 31normal conjunctival samples, Huang et al. identified a high microbial diversity, classified into 25 phyla and 526 distinct genera, providing a framework to investigate the potential roles played by diverse microbiota [10]. In terms of composition, these studies concluded that Proteobacteria was the most abundant phylum on the normal conjunctiva while Actinobacteria and Firmicutes were the next two most abundant phyla [9, 10].”

  • The authors have been described the sample collection and selection. Provide some detail of durations between sampling and samples processing.

Thank you for your comment. In section 2.2, we added this sentence in the last paragraph to clarify this point as per your suggestion:” Specimens were collected and the initial conventional cultures (blood and a chocolate agar nutrient medium) were conducted at Khanh Hoa General Hospital same day of sampling. Then ocular samples were stored at -20 °C until transferred to our hospital. DNA was extracted and screened by real-time PCR and 16S rRNA sequencing assay in the Department of Laboratory Medicine at Nagasaki University Hospital on March 2018.”

  • Provide detailed metadata including Age, sex, uni/bilateral, with/without antibiotic, date of sample collection, processing info – immediately or stored for sample pool.

We added Table (1) including detailed metadata as per your suggestion.

  • Could the authors comment on their choice to use the OTU approach rather than the recently more commonly used ASV approach?

Thank you for your meaningful comments.

In our understanding, ASV has been generally available on QIIME2 pipeline. As you suggested, we are also interested in ASV approach because noise reduction algorithms will contribute to our understanding the bacteria more deeply. However, we are not ready to use it because we use CLC-workbench, a commercial analysis software, as a standard analysis tool in order to support other many genetic analyses in our hospital.

In our consideration, our work presented and discussed data up to the Genus level because the OTU approach can have uncertain data at the Species level.

Thus, it is difficult to use it at this time but we would like to use ASV if it becomes available in our lab.

  • The authors have provided the Shannon index for alpha diversity, however would like to see the relative abundance of the bacterial diversity for each sample. So I would suggest the author provide the relative abundance (Figure 5 A and C) of all 47 samples (both genus and phylum level) and discuss how the evenness and richness of the diversity using the metadata.

We added figures about the relative abundance of all samples (as Figure 5) and the new section in the results (as 3.3). Figures for genus level were made with only major genera because 482 genera were detected in our study. We also evaluated the evenness and the richness for genus and phylum level (please see below). However, because there are little findings and we already presented the figures about Linear Discriminant Analysis, we thought it would be better to present only the new Figure 5 to avoid confusing.

In addition, we added a new section (3.3) titled with (Relative abundance of each sample). The new section will appear in the manuscript as follow:

3.3. Relative abundance of each sample

            Next, the relative abundance of each sample at phylum and genus levels were analyzed and compared based on the information about the use of antibiotics and laterality. In the analysis of the phylum level, significant findings were not observed between the groups for the use of antibiotics (Figure 5A), however, for the laterality, samples in the bilateral group seemed to be relatively rich in Firmicutes rather than the unilateral group (Figure 5B). In the analysis of representative genera (which had at least 50,000 reads in this study), Streptococcus, Cutibacterium, and Staphylococcus genera were observed in most of samples. Samples with unique genera with high abundance, such as Porphylomonas, Haemophilus, Corynebacterium, Acinetobacter, Janibacter, Anaerococcus were also observed (Figure 5C, 5D).”

The previous section will be followed with section 3.4. Linear Discriminant Analysis:

  • No detailed report on the Beta diversity authors should provide more detail on this aspect.

Thank you for the comment. We added some sentences in the second paragraph of 4.2. Microbial diversity section. The paragraph will be as follow:” In our study, there were no difference in Beta diversity analysis for sex (P=0.087), chemosis (P=0.064), unclassified eye drops (P=0.431), pain (P=0.315) and itching (P=0.133). There was a statistical significant difference between bilateral versus unilateral conjunctivitis (P=0.017) and for antibiotics (P=0.020) (Figure 3A, B), although there were no statistical significant differences in these factors using Shannon index (Alpha diversity analysis). These findings suggested that the bacterial composition of the microbiota was changed without reducing the species diversity. Alteration of the microbiota due to antibiotics usage wass characterized with the reduction of some susceptible bacteria and the induction of microbial substitution. In addition, it was confirmed that antibiotic administered prophylactically before ophthalmic procedures may further reduce the ocular surface’s amount and variety of microbiota [21, 22]. Reasonably we can accept that, bacterial composition of the microbiota in bilateral cases were different from unilateral cases. However, the stability in the alpha diversity can be a characteristic of the ocular microbiota in contrast to skin microbiota [23]. These results, to the best of our knowledge, are considered to be the first diversity analyses for subjects with conjunctivitis reported in the literature.”

  • Figure 6 color pattern unable to distinguish so use different colors.

We changed the Figure 6 color pattern as per your suggestion (becomes Figure 7).

  • Authors claiming this is the first study trying to identify bacterial composition in human conjunctivitis but few investigations have been listed by authors and few pieces of literature available. I would suggest rephrasing or remove the sentence.

We removed this sentence as per your suggestion.

NB: The manuscript was sent for English revision as per your suggestion.

Reviewer 2 Report

The authors of the present manuscript examined the microbiota via conventional culture, PCR and 16S ribosomal sequencing of 50 eyes collected over about a 1 year period with bacterial conjunctivitis. The results are interesting and valuable to the scientific and clinical community; however, the study does have major limitations. Below are my comments.

Major:

1.) The authors appropriately acknowledged the major limitation to the study in that they do not have any control samples. It would have been helpful to have the controls because one of the authors conclusions is that pseudomonas is a protective factor when present in the normal microbiota of the ocular surface, which is not present in the current study. This does not enable the authors to support this conclusion albeit other studies have made similar conclusions. The absence of pseudomonas could be argued to be due to methodologic error in the present study. It would have also been interesting to use the unaffected eye as its own control in unilateral cases. 

2.) My other major concern is the power of the study. Do the authors believe that there is insufficient power with only 47 of 793 samples examined? I understand the financial limitations being the driving force of this, but I wonder if some of the findings were not statistically significant due to too few samples, such as vision loss, sex, chemosis that were all trending towards significance. 

Minor concerns:

1.) abstract really should clarify that only 47 samples collected over the time frame were examined.

2.) Introduction - would be helpful to cite typical microbiota composition in healthy eyes order to compare the present study's  findings.

3.) section 2.1 - last sentence - just use the word "associated" once in the sentence.

4.) Section 2.2 - were all of the infections universally severe? Seems unusual that every case of bacterial conjuncitivits would be "severe". What objectively determines if the infection is severe?

5.) PCR primers should be provided by the authors.

6.)  section 3.4 - in case No.401, the  . . . should be "bacterium"

7.) Section 4.1 - try to combine the 1-sentence paragraphs. 

8.) Section 4.1 - paragraph 4 - this is an incomplete sentence / grammatically incorrect.

Author Response

18-June-2021

Thank you very much for your comments on our manuscript.

We have revised the manuscript as per your suggestion.

Reviewer #2:

Comments to the Author:

Major:

  • The authors appropriately acknowledged the major limitation to the study in that they do not have any control samples. It would have been helpful to have the controls because one of the authors conclusions is that pseudomonas is a protective factor when present in the normal microbiota of the ocular surface, which is not present in the current study. This does not enable the authors to support this conclusion albeit other studies have made similar conclusions. The absence of pseudomonas could be argued to be due to methodologic error in the present study. It would have also been interesting to use the unaffected eye as its own control in unilateral cases. 

Thank you for your comment. We changed the sentence in Discussion, section 4.1: “In the present study, it was confirmed that Pseudomonas was present in a scant percentage of cases of conjunctivitis which may agree with previous study in their suggestion about the importance of Pseudomonas as resident commensals microbiota for prevention of infection. Further study is needed to confirm this conclusion with own control specimens.” as per your suggestion.

Also in Conclusion and Abstract, we changed the sentence:”Of the predominant phyla, Firmicutes had the highest abundance in bacterial conjunctivitis in this study.  Pseudomonas as a resident commensal microbiota may have an important role in the prevention of infection.” As per your suggestion.

  • My other major concern is the power of the study. Do the authors believe that there is insufficient power with only 47 of 793 samples examined? I understand the financial limitations being the driving force of this, but I wonder if some of the findings were not statistically significant due to too few samples, such as vision loss, sex, chemosis that were all trending towards significance. 

Thank you for your comment. We agreed with you that financial limitation considered one of driving forces of this number of samples. In addition, our study number of samples examined is not too few in comparison to other papers describing ocular surface microbiome in allergic conjunctivitis, Steven Johnson syndrome, fungal keratitis, dry eye, bacterial keratitis, and in diabetics etc….

Minor:

  • abstract really should clarify that only 47 samples collected over the time frame were examined.

We changed the sentence in abstract as per your suggestion:” The study included 47 randomly selected patients.”

  • Introduction - would be helpful to cite typical microbiota composition in healthy eyes order to compare the present study's findings.

 We added the following sentences in Introduction as per your suggestion.

“In 2011, a first pilot study involving 4 normal subjects was conducted with an aim to explore the bacterial diversity of a healthy human conjunctiva using 16S rRNA sequencing [9]. The study revealed an unpredicted diverse microbial community and identified that healthy conjunctival microbiome was dominated by Proteobacteria, Actinobacteria and Firmicutes bacterial phyla. The most common taxa at the genus level were Pseudomonas, Propionibacterium, Bradyrhizobium, Corynebacterium, Acinetobacter, Brevundimonas, Staphylococci, Aquabacterium, Sphingomonas, Streptococcus, Streptophyta and Methylobacterium [9]. In 2016, using the same technology to analyze 31normal conjunctival samples, Huang et al. identified a high microbial diversity, classified into 25 phyla and 526 distinct genera, providing a framework to investigate the potential roles played by diverse microbiota [10]. In terms of composition, these studies concluded that Proteobacteria was the most abundant phylum on the normal conjunctiva while Actinobacteria and Firmicutes were the next two most abundant phyla [9, 10].”

  • section 2.1 - last sentence - just use the word "associated" once in the sentence.

 We changed the sentence in section 2.1 as per your suggestion:” Clinical-epidemiological data includes age, sex, laterality of infection, associated chemosis, pain, itching, visual loss, usage of unclassified eye drops, and antibiotic eye drops.”

  • Section 2.2 - were all of the infections universally severe? Seems unusual that every case of bacterial conjuncitivits would be "severe". What objectively determines if the infection is severe?

    Thank you for your comment. We replaced word “severely” with “more” because it is misleading one. So, we changed the sentence into:” Conjunctival swabs were obtained carefully from the more infected patient’s eye.”    

  • PCR primers should be provided by the authors.

We added PCR primers as a supplementary file 1 at the end of 2.4. Section. 

  • section 3.4 - in case No.401, the  . . . should be "bacterium"

 We changed the word as per your suggestion.

NB: Section 3.4 becomes 3.5

  • Section 4.1 - try to combine the 1-sentence paragraphs. 

    We removed the first sentence in this section as per one reviewer suggestion. So, the first paragraph will begin with this sentence:”A culture-independent approach by 16S rRNA metagenome sequencing has been used to identify our microbiota during both health and illness.”

  • Section 4.1 - paragraph 4 - this is an incomplete sentence / grammatically incorrect.

     We corrected the paragraph after English review as per your suggestion. The paragraph is changed into:” Although Firmicutes was the least abundant in their studies among the main first three phyla (Proteobacteria, Actinobacteria, Firmicutes), in the present study it had the highest abundance (one of the most significant results in this study).”

NB: The manuscript was sent for English revision as per your suggestion.

Round 2

Reviewer 1 Report

The authors have addressed all my questions and provided details as per my suggestions. 

Reviewer 2 Report

I'd like to thank the authors for their thorough and well-thought reply and revisions. They have adequately addressed my concerns and I believe the information they provide in the present manuscript is important to the scientific and clinical communities; therefore I recommend it for publication.